Intelligent sensors in assistive systems for deaf people: a comprehensive review

Sabino Soares Caio César 1
Silva Luis Augusto luisaugustos@usal.es 2
Fernandes Anita 1
Villarrubia González Gabriel 2
Leithardt Valderi R.Q. 3
Delcio Parreira Wemerson wemerson.delcio@puc-campinas.edu.br 4
1 Polytechnic School, Universidade do Vale do Itajaí , Itajai , Santa Catarina , Brazil
2 Department of Computer Science, Universidad de Salamanca , Salamanca , Spain
3 Instituto Universitário de Lisboa (ISCTE-IUL), ISTAR , Lisboa , Portugal
4 Faculty of Electrical Engineering, Polytechnic School, Pontifical Catholic University of Campinas , Campinas , São Paulo , Brazil
Comai Sara
Electronic publication date: 2024 Oct 24
Publication date: 2024
Volume: 10
Electronic Location ID: e2411
Received 2024 Jul 15; Accepted 2024 Sep 23
Copyright: ©2024 Sabino Soares et al.
Copyright year: 2024
Copyright holder: Sabino Soares et al.
License: This is an open access article distributed under the terms of the Creative Commons Attribution License, which permits unrestricted use, distribution, reproduction and adaptation in any medium and for any purpose provided that it is properly attributed. For attribution, the original author(s), title, publication source (PeerJ Computer Science) and either DOI or URL of the article must be cited.
License URL: https://creativecommons.org/licenses/by/4.0/

Keywords: Smart sensors, Internet of Things (IoT), Assistive devices, Deaf and hard of hearing, Communication enhancement

Funding: The project Self-adaptive platform based on intelligent agents for the optimization and management of operational processes in logistic warehouses (PLAUTON) PID2023-151701OB-C21 MCIN/AEI/10.13039/501100011033/FEDER The APC was supported by the project Self-adaptive platform based on intelligent agents for the optimization and management of operational processes in logistic warehouses (PLAUTON) PID2023-151701OB-C21, funded by MCIN/AEI/10.13039/501100011033/FEDER, EU. The funders had no role in study design, data collection and analysis, decision to publish, or preparation of the manuscript.

==============================
This research aims to conduct a systematic literature review (SLR) on intelligent sensors and the Internet of Things (IoT) in assistive devices for the deaf and hard of hearing. This study analyzes the current state and promise of intelligent sensors in improving the daily lives of those with hearing impairments, addressing the critical need for improved communication and environmental interaction. We investigate the functionality, integration, and use of sensor technologies in assistive devices, assessing their impact on autonomy and quality of life. The key findings show that many sensor-based applications, including vibration detection, ambient sound recognition, and signal processing, lead to more effective and intuitive user experiences. The study emphasizes the importance of energy efficiency, cost-effectiveness, and user-centric design in developing accessible and sustainable assistive solutions. Moreover, it discusses the challenges and future directions in scaling these technologies for widespread adoption, considering the varying needs and preferences of the end-users. Finally, the study advocates for continual innovation and interdisciplinary collaboration in advancing assistive technologies. It highlights the importance of IoT and intelligent sensors in fostering a more inclusive and empowered environment for the deaf and hard-of-hearing people. This review covers studies published between 2011 and 2024, highlighting advances in sensor technologies for assistive systems in this timeframe.

Introduction

According to the World Health Organization WHO (2022), hearing impairment represents a significant public health challenge affecting approximately 1.5 billion individuals globally. This number is anticipated to increase to around 2 billion by 2050 if effective preventive and treatment strategies are not adopted. Among the affected, 34 million are children who have deafness or hearing loss, with 60% of these instances deemed preventable through appropriate interventions. The economic impact of untreated hearing loss is profound, with an estimated annual global cost of US $980 billion.

To enhance the quality of life for those with hearing impairments, assistive technologies (AT) play a pivotal role. AT encompasses a multidisciplinary field that includes products, resources, methodologies, strategies, practices, and services to foster autonomy, independence, quality of life, and social inclusion for individuals with disabilities (Rohwerder, 2018).

Prominent among the ATs for individuals who are deaf or hard of hearing are hearing aids and cochlear implants, which amplify and process sounds to facilitate auditory perception (Do Carmo, Borsoi & Costa, 2022; Borges, Parreira & Costa, 2019a; Borges, Parreira & Costa, 2019b; Souza, Costa & Borges, 2024). Moreover, current research explores the potential of communication technologies and the Internet of Things (IoT) to improve the interaction of those who are deaf or hard of hearing with their environment. This involves leveraging smart sensors to capture and transmit relevant information (Santhalingam et al., 2020; Liu, Jiang & Gowda, 2020; Lee, Chong & Chung, 2020; Kim et al., 2020).

The communication barrier significantly undermines the autonomy and freedom of individuals with hearing loss, necessitating the support of specialized professionals. However, many consequences of hearing loss can be mitigated through early detection and intervention. Such measures include specialized education programs, sign language instruction for children and their families, assistive technologies like hearing aids and cochlear implants, and speech therapy and auditory rehabilitation services. These interventions aim at achieving equality, accessibility, and effective communication for the deaf and hard of hearing (Rodríguez-Correa et al., 2023; Vogel & Korte, 2024).

The focus on sensors in this review is justified by their central role in modern assistive systems for individuals with hearing impairments. Sensors are crucial for capturing and interpreting environmental signals, enabling the integration of various stimuli (Souza, Costa & Borges, 2024), such as sound and motion, into accessible assistive devices. Sensor technologies, including accelerometers, high-precision cameras, and microphones, can detect and convert visual and auditory information into processable signals by intelligent systems, providing a more efficient and personalized interface for users. Recent studies, such as Lavric et al. (2024), Manzoor et al. (2024), and Pydala, Kumar & Baseer (2024), highlight the increasing use of sensors to enhance the quality of life for deaf individuals by enabling real-time interaction with their surroundings. Furthermore, the flexibility of sensors to be integrated into wearable technologies amplifies their application in assistive solutions beyond communication, extending to environmental navigation and alert signal perception (Pramod, 2023; Aashritha & Manikandan, 2023). Therefore, emphasizing sensors not only reflects the latest technological innovations but also underscores their potential to provide greater autonomy and social inclusion for people with hearing impairments.

This work presents a systematic literature review (SLR) to identify how intelligent sensors are employed in assistive technologies for the deaf and hard of hearing. It explores the most commonly used sensor types, communication protocols, costs, and feasibility. This SLR seeks to answer specific research questions to understand these technologies’ impact and utility thoroughly.

(i) What are the main functions of the sensors found applied to ATs?

(ii) What is the availability and access to the sensor?

(iii) How much electrical energy do the sensors consume to operate?

(iv) What are the communication protocols of the sensors?

(v) Do the sensors require constant maintenance?

(vi) What challenges are described in the studies?

(vii) What are the physical proportions of the sensors?

(viii) Can the sensors be used individually?

This article is structured as follows. Section ‘Background’ has the theoretical foundation, which addresses the concepts presented, such as smart sensors and communication through the Internet of Things. Section ‘Systematic Literature Review’ describes the methodology applied in this systematic review, including the planning, keywords, inclusion and exclusion criteria, search databases, data extraction, and charts. Subsequently, Section ‘Results and Discussions’ presents the research results, discussions, and comparative tables. Finally, Section ‘Conclusions’ provides a conclusion, followed by the acknowledgments and bibliographic references.

Background

The communication barrier significantly underscores the challenges individuals with hearing loss face, particularly, their diminished autonomy and restricted ability to perform daily tasks independently. This often necessitates support from trained professionals. However, the adverse effects of hearing loss can be substantially reduced through early detection and timely interventions (World Health Organization, 2022). Effective assistive technologies, including hearing aids, cochlear implants, visual alert systems, and various mobility aids such as wheelchairs and canes, are crucial in supporting individuals of all ages with hearing loss (Soetan et al., 2020). Additionally, services like speech therapy and auditory rehabilitation enhance equality, accessibility, and communication, improving these individuals’ overall quality of life.

Assistive technology can be defined as any product primarily designed to maintain or enhance an individual’s functioning and independence, thus promoting their well-being. These technologies can be specifically produced or available on the market for the general public and can be used preventively in the face of a particular disability (Rohwerder, 2018).

However, these assistive technologies are often abandoned by users (Rohwerder, 2018). The reasons are complex but centre on a mismatch between the user’s need and the device’s ability to perform its intended function. People often purchase a particular technology because it’s more feasible to obtain, even if it doesn’t best meet their needs; another scenario arises when the user’s needs change, rendering the technology outdated. The need for technologies that adapt to user demands is clear. However, this situation has progressively evolved, and currently, there’s a paradigm shift in design where the user is at the centre of creation, involved in manufacturing processes as a kind of “co-creator” (Rohwerder, 2018). This factor, combined with the emergence of sensing modalities, has revolutionized how disability is addressed in society.

Smart sensors

A sensor is a device that converts a physical quantity into a corresponding electrical signal, which can be either analog or digital. Digital sensors provide significant advantages over analog sensors, including advanced signal conditioning, built-in analog-to-digital conversion, and sophisticated data processing capabilities. These features impart a certain level of intelligence to digital sensors, allowing them to perform more complex tasks (Song, FitzPatrick & Lee, 2017).

Smart Sensors elevate this concept by incorporating microprocessors within digital sensors, enabling them to execute logical functions and make decisions autonomously. Using a communication interface, they are engineered to process and interpret data from various sensors and communicate it to users through a bidirectional digital bus. This integration transforms them into multifunctional devices capable of handling many applications (Middelhoek & Hoogerwerf, 1985; Song, FitzPatrick & Lee, 2017).

The architecture of Smart Sensors is based on three core modules: the detection module, the processing module, and the network communication module (Song et al., 2022). The detection module includes an array of sensors that capture physical data. The processing module, divided into four submodules—time and synchronization, signal processing, data processing, and metadata management—enables the sensor to process and analyze the captured data. The network communication module ensures seamless communication through standard protocols, facilitating the integration of sensors into broader intelligent networks. These capabilities allow Smart Sensors to feature advanced functionalities such as time synchronization, self-identification, and self-description, enhancing their utility in various applications (Song, FitzPatrick & Lee, 2017).

In essence, Smart Sensors possess four primary capabilities: (1) sensing the environment, (2) performing analog signal conditioning and analog-to-digital conversion (ADC) within the module, (3) maintaining precise timing and synchronization via an internal clock, potentially supplemented by an external time reference, and (4) conducting network communications through a dedicated module. These sensors convert physical phenomena into electrical signals, such as measuring voltage and current in power lines of microgrids. They utilize various communication protocols to interact with external systems, including the IEEE 1451 family of Smart Transducer Interface Standards, IEEE 1815 Standard for Electric Power Systems Communications (DNP3), and IEEE C37.238 PTP Power Profile, among others (Song, FitzPatrick & Lee, 2017).

Furthermore, the implementation of Smart Sensors in various fields highlights their versatility and adaptability. In industrial automation, they monitor and control machinery operations, enhancing efficiency and reducing downtime. Smart Sensors track parameters like temperature, humidity, and pollution levels in environmental monitoring, providing critical data for climate studies and disaster management. In healthcare, these sensors enable remote patient monitoring, collecting vital signs and transmitting data to healthcare providers for real-time analysis and intervention. The ability of Smart Sensors to integrate with Internet of Things (IoT) networks further amplifies their potential, allowing for real-time data collection, processing, and communication across diverse applications.

Smart Sensors are also pivotal in advancing smart city initiatives, contributing to intelligent traffic management systems, energy-efficient building operations, and enhanced public safety measures. Their role in developing autonomous vehicles is crucial, as they provide the necessary data for navigation, obstacle detection, and decision-making processes. The continuous evolution of Smart Sensor technology promises even more innovative applications, driven by advancements in microelectronics, data analytics, and wireless communication.

Smart Sensors are also pivotal in advancing smart city initiatives, contributing to intelligent traffic management systems, energy-efficient building operations, and enhanced public safety measures. Their role in developing autonomous vehicles is crucial, as they provide the necessary data for navigation, obstacle detection, and decision-making processes. The continuous evolution of Smart Sensor technology promises even more innovative applications, driven by advancements in microelectronics, data analytics, and wireless communication. Their ability to sense, process, and communicate data autonomously makes them indispensable in modern technological landscapes. Their integration into various sectors improves operational efficiency and opens new avenues for innovation and development (Song, FitzPatrick & Lee, 2017).

IoT communication

With the rapid development of the Internet and wired and wireless communication technologies, the popularity of the Internet of Things (IoT) has witnessed a drastic increase in its daily use. Moreover, emerging technologies, such as radio frequency identification and tracking, intelligent sensors, and industrial and environmental detection networks, make connecting physical things or objects on an IoT network possible. Due to this technology’s characteristic connectivity to anything, anytime, and anywhere, it is being extensively used in various applications worldwide (Khang, Yu & Lee, 2015; Da Xu, He & Li, 2014).

IoT was initially proposed to refer to connected and interoperable objects uniquely identifiable by radio frequency (RFID). Later, researchers related IoT to more technologies, such as sensors, actuators, GPS devices, and mobile devices (Da Xu, He & Li, 2014).

Today, a commonly accepted definition for IoT is a dynamic global network infrastructure with self-configuring capabilities based on standard and interoperable communication protocols. Physical and virtual things have identities, physical attributes, and personalities, using intelligent interfaces seamlessly integrated into the information network (Van Kranenburg, 2007).

Specifically, integrating sensors and actuators, RFID tags, and communication technologies serve as the foundation of IoT and explain how various objects and physical devices around us can be associated with the Internet. Thus, these objects and devices can cooperate and communicate with each other to achieve common goals (Van Kranenburg, 2007).

IoT is a global network infrastructure composed of several connected devices that rely on sensory, communication, network, and information processing technologies (Tan & Wang, 2010). A fundamental technology for IoT is RFID technology—already mentioned—which allows microchips to transmit identification information to a reader through wireless communication. By using RFID readers, people can identify, track, and monitor any objects attached to RFID tags automatically (Jia et al., 2012). Another fundamental technology for IoT is wireless sensor networks (WSNs), which mainly use interconnected intelligent sensors to detect and monitor (Da Xu, He & Li, 2014). Advances in RFID and WSNs provide a significant contribution to the development of IoT. Moreover, many other technologies and devices are used to form an extensive network that supports IoT.

As more devices are equipped with intelligent technology, connecting things becomes easier. In the sensing layer, wireless smart systems with tags or sensors can now detect and exchange information automatically between different devices (Da Xu, He & Li, 2014). These technological advances significantly improve IoT’s ability to detect and identify things or environments.

The Internet of Things has some fundamental layers for its operation, such as the network, service, and interface layers, but what interests us here are communication technologies. This network can contain many electronic devices, mobile devices, and industrial equipment. Different things have different parameters. This scenario involves a series of heterogeneous networks, such as WSNs, wireless mesh networks, and WLAN, all of which assist in information exchange (Da Xu, He & Li, 2014). In this case, a gateway can facilitate communication or interaction of various devices over the Internet. This same service can also be taken advantage of through the network running local optimization algorithms to deal with the vast amount of data generated by the IoT (Zhu et al., 2010).

The main communication protocols and standards include RFID (e.g., ISO 18000 6c EPC class 1 Gen2), NFC, IEEE 802.11 (WLAN), IEEE 802.15.4 (ZigBee), IEEE 802.15.1 (Bluetooth), Multihop Wireless Sensor/Mesh Networks, IETF Low Power Wireless Personal Networks (6LoWPAN), machine to machine (M2M) and traditional IP technologies, such as IP, IPv6, etc. There are also some cross-layer protocols for wireless networks, such as wireless sensor and actuator networks (WSANs) or ad hoc networks (AHNs) (Han et al., 2013).

Systematic Literature Review

A fundamental part of this work, the systematic literature review, is identifying, evaluating, and interpreting all available research relevant to a particular research question, thematic area, or phenomenon of interest. The individual studies that contribute to a systematic review are called primary studies; a systematic review is a form of secondary study (Kitchenham, 2004; Khan & Keung, 2016).

According to Kitchenham et al. (2009), SLR is the systematic way of reporting outcomes extracted from the literature. The SLR method offers a way to classify, explore, and examine contemporary studies related to any questions of interest. The author classifies the SLR into three phases: planning, conducting, and reporting the review. The SLR protocol is, therefore, the result of the planning phase.

Methodology

The methodology used was based on planning, conducting, and reporting, which was mentioned in the previous topic by the author, Kitchenham, but not necessarily because of him. For the organization of this RSL, the Parsifal platform (Perform Systematic Literature Reviews) was used, which has features that facilitate and speed up the extensive work of the researcher during this process. It was beneficial for organization and information sharing, aligning all stages automatically and bringing the desired results much more quickly. This platform has a clear and objective division of the three stages, which we will address individually.

Planning

In this bibliographic survey, we used a rigorous and reproducible procedure. In the planning part, a protocol was used (identification of the research objective, elaboration of the PICOC, definition of research questions, keywords and synonyms, search string, research bases, selection, and exclusion criteria), a quality assessment checklist (questions, answers, and scores), and the data extraction form.

In the protocol, the research objective is clear and is defined right at the beginning: “Select the most suitable smart sensors for assistive technologies for deaf people”. Based on this, the other stages of planning this RSL were woven. Regarding the elaboration of the PICOC, this process took the form described in Table 1.

The research questions, crucial for completing the Systematic Literature Review (SLR), were answered using the data extracted from the research and are displayed in Table 2.

The keywords and synonyms were a constituent part of the search string, essential for carrying out the research phase and are displayed in Table 3.

This set of keywords generated the crucial part for conducting a targeted search: the search string. In this case, it took the following format: “(“deafness” OR “hard to hear” OR “hearing impaired” OR “hearing impairment” OR “sensorineural loss” OR “conductive loss”) AND (“assistive technology”) AND (“sensor” OR “intelligent sensor”) AND (“IoT” OR “internet of things”)”. It was used in the research databases in Table 4.

To finalize the protocol stage, the step determines which article remains based on the inclusion criteria from Table 5. It also determines which will be discarded based on the exclusion criteria from Table 6. The selection criteria were designed to play a decisive role in selecting articles.

Table 1 Population, intervention, comparison, outcomes, and context (PICOC).

PICOC	Description	
Population	Hearing-impaired individuals	
Intervention	Assistive technology	
Comparison	Systematic Literature Review (SLR), systematic mapping	
Outcomes	Sensors, hardware, protocols	
Context	Primary studies, IoT, Internet of Things	

Table 2 Research questions.

Order	Questions	
1	What is the function of the sensor?	
2	What is the availability and access to the sensor?	
3	How much electrical energy does the sensor consume to operate?	
4	What are the communication protocols of the sensor?	
5	Does the sensor require constant maintenance?	
6	What challenges are described in the study?	
7	What are the physical proportions of the sensor?	
8	Can the sensor be employed individually?	

Table 3 Keywords & synonyms.

Keyword	Synonyms	Related to	
IoT	Internet of things	Outcome	
Assistive technology	–	Intervention	
Deafness	Hard to hear, hearing impaired	Population	
Sensor	Intelligent sensor	Comparison	
Sensorineural loss	Conductive loss	Population	

With the protocol stage defined, but still in the Planning phase, there is the quality assessment checklist, in Table 7, which contains the research questions and their weights, found in Table 8.

Table 4 Research sources.

Sources	URL	
ACM Digital Library	http://dl.acm.org/	
Begell Digital	http://www.dl.begellhouse.com/	
Engineering Village	https://www.engineeringvillage.com/	
IEEE Xplore Digital Library	http://ieeexplore.ieee.org/	
Journal Citation Reports	https://jcr.clarivate.com/	
MathSciNet®	http://www.ams.org/mathscinet/	
PubMed	http://www.ncbi.nlm.nih.gov/pubmed	
ScienceDirect Database	http://www.sciencedirect.com/	
Scopus	http://www.scopus.com	
Springer	http://www.springer.com/	
The Lancet	https://www.thelancet.com/	
Web of Science	https://webofknowledge.com/	

Table 5 Inclusion criteria.

List of inclusion criteria	
Study that addresses sensors in assistive technology(ies) for hearing impairment.	

Table 6 Exclusion criteria.

List of exclusion criteria	
Study outside the scope/area (intelligent sensors most suitable for assistive technologies for deaf people)	
Inaccessible, unavailable study, without free access	
The study does not present the technical specifications of the sensor	
A study published in journals ranked in Q3 and Q4 quartiles of SJR	
Secondary studies (meta-analyses, surveys, SLR, and literature reviews) or tertiary (review of reviews)	

Table 7 Quality assessment checklist.

No	Quality assessment	
1	Do the authors describe the limitations of the study?	
2	Are the technical criteria of the sensors well described?	
3	Is the research objective clearly described and defined?	
4	Does the study identify intelligent sensors aimed at hearing impairments?	
5	Did the study conduct a well-described experiment to evaluate the proposal?	
6	Is the study cited by other authors?	

Table 8 Answers and weights.

Description	Weight	
Yes	1.0	
Partially	0.5	
No	0.0	

The last step in the protocol process is the Data Extraction Form, shown in Table 9, indicating which information should be present in each article and what type they are.

Table 9 Data extraction form.

Description	Type	Values	
Authors	String Field	n/a	
Publication date	Date Field	n/a	
Country of publication	String Field	n/a	
Implementation requirements of the identified sensor	Boolean Field	n/a	
Protocol used in the sensor	Select Many Field	?	

Conduct

With the well-defined protocol stage, it was possible to establish the foundations for starting the research. The CAPES Periodicals Portal was used to locate the previously defined research databases. Access was made through institutional login at the Federated Academic Community (CAFe).

Locating the databases in the Portal was straightforward. In some cases, the challenge was using the search mechanisms of the platforms. Initially, 12 databases were chosen to conduct the search using the string. However, with the first round of searches in these databases–constituting the first stage of the Conduct process–it was observed that the Begell Digital, Journal Citation Report, MathSciNet®, and The Lancet databases presented numerous difficulties in using their mechanisms. Moreover, they provided few relevant results. In the second round of searches, they were still maintained for tests, but in the third round, these databases were eliminated. In the end, there were eight databases in total.

As mentioned, several rounds of research were conducted in the selected databases to check what beneficial results they returned based on the search string determined in the Protocol stage. It is essential to mention that it was necessary to make minor adjustments to this string to fit the format of the chosen periodical database. In summary, the periodicals observed in Fig. 1 were obtained at the end of the search rounds.

A significant increase was observed over the years in the number of articles related to the research topic in the initial inclusion, as shown in Fig. 2. This may indicate a growing trend in the research area of assistive technologies for the deaf over the years.

The 781 journals were exported directly from the suggested databases in the BibTeX file format, making it possible to directly upload them to the Parsifal tool. Before proceeding at this stage, a process of checking for duplicate journals was carried out, as in some instances, the same journal appeared in multiple databases. In this procedure, which is automated by the Parsifal tool itself after indicating that you wish to perform the check, a total of 65 journals were categorized as duplicates and then removed. At the end of the procedure, there were 716 documents for future analysis.

With the inclusion stage of the journals in the tool, a visualization panel was available showing all the journals. In addition, there was a set of relevant information about it, such as title, authors, publication year, journal in which it was published, keywords, BibTex Key, document type, DOI, language, and abstract. To start the process of analyzing these journals, that is, to decide which one will be approved or rejected, the abstract, title, document type, and complete consultation of the journal through the DOI were mainly analyzed. With this information, in most cases, it was already possible to determine whether the journal met the requirements or was out of the desired scope.

Figure 1 Journals accepted per year.

Figure 2 Total number of articles per year.

Furthermore, in cases where the article passed the initial checks, it was necessary to consult the journal’s quartile to determine its relevance and then reject or accept it. Thanks to the use of the Parsifal tool, this step was extremely practical, as with just a few clicks, it was possible to “judge” a particular journal and then move on to the next one. Thus, this stage was concluded, which, although it was the longest, was completed quickly considering the volume of journals.

The inclusion and rejection process was governed by a lot of reading and verification to ensure the origin of a particular journal. In some cases, discarding the journal in a few seconds was possible, as it was part of a survey. Therefore, it fell under the exclusion criteria. In others, the process was more difficult due to the lack of information contained in the file exported from the journal database. It was necessary to consult the journal and read it more carefully to finally understand what that article was about, where it was published, and its relevance to the academic community. At times, a particular journal seemed to fit perfectly into what was being sought, but due to some details, it was rejected at the end of the analysis. On the other hand, some rejected journals ended up being accepted after their information was checked again with greater attention.

Given this, at the end of this back-and-forth process, a preliminary result of 17 journals was obtained—most from 2022, as shown in Fig. 3—to be analyzed individually. We will primarily conduct an assessment of the source and its content, address the research inquiries, and ascertain its suitability for retention or disposal.

Results and Discussions

Reading the articles was a lengthy and meticulous process, given the importance of this stage in the development of this work. The dynamics worked as follows: the articles were downloaded in Portable Document Format (PDF) directly from the database to which they belonged. They were organized in a cloud folder - using the OneDrive tool from Microsoft in association with the institutional email - allowing reading to take place anywhere, anytime. From this, an Excel control spreadsheet was created, containing information such as name, objective, problem, sensors, communication protocols, applications, techniques, and methodological gaps. As the files were read, this table was filled in parallel with the information in the Parsifal tool.

The biggest difficulty encountered at this stage was the language. Since all the articles were in a foreign language, certain parts of the text seemed strange due to a lack of proficiency. To solve the problem, translation tools such as Google Translate and DeepL were used to decipher some sentences. However, the translation using these tools, although they perform their task very well in most cases, still cannot translate specific contexts as expected. This generated some ambiguity or took away the meaning of certain words.

Figure 3 Articles extracted from the search string in the databases indicated.

Next, the situation of each article will be described after reading its entire content. It is worth noting that there was a satisfactory variation regarding each research’s sensors and final products. The solutions ranged from the use of wearable equipment, such as gloves, rings, bracelets, bracelets, and lenses to sensors that go through augmented reality. It was disappointing that many articles brought a very interesting solution to the problem that this Systematic Review seeks to solve, but they failed to describe the sensors, their technical specifications, cost, and communication protocols.

Of the initial 17 articles, four were discarded during reading. The first, called “An Overview of the Internet of Things for People with Disabilities”, was removed from the list of articles of interest once its revisionary nature was verified. This is unsuitable for this Systematic Review, primarily because it addresses the theme of sensors very little. It superficially provides an overview of what is available today in IoT, its architecture, some communication protocols, the market, the benefits of using this technology, and some results of recent work in this area. It is interesting, but outside of what is sought here.

The other article that was rejected post-reading was “Envisioning Inclusive Futures-Technology-based Assistive Sensory and Action Substitution”. It shows the reader the possibilities of how a connected world and growing technological evolution can positively change the lives of people with disabilities. It addresses what is currently trending, the prospects for changes in this area, what is available regarding accessibility in modern devices, and related things. However, it addresses very few technical requirements and protocols, making it insufficient to extract its content and use it for the purpose of this research.

The article called “Synergistic Integration between the Internet of Things and Augmented Reality Technologies for deaf people in E-learning Platform” was left out because its content, although addressing a solution for the hearing impaired, did not address the technical specifications of sensors. The article provides a solution in the field of education, with teaching applied to augmented reality technologies and active voice interpretation, but fails to provide the means and methods used, such as the structure and technological apparatus in technical lines.

Finally, “Ultralight Iontronic Triboelectric Mechanoreceptor with High Specific Outputs for Epidermal Electronics” was left behind due to its complex understanding and solution that, although applicable to the target audience of this research, did not propose to develop a straightforward and directed synthesis. The article addresses more of a health monitoring solution but does not bring something innovative and specifically applicable to the hearing impaired.

Functions

Intelligent sensors are devices that can capture, process, and transmit data from their surroundings using internal computing resources. They can be applied to assistive technologies to facilitate the lives of people with disabilities, incapacities, or limitations, as previously seen. Throughout the reading of the article, the following sensor categories were identified: measurement, detection, monitoring, tracking, and recognition.

Intelligent sensors can measure physical or chemical variables, such as temperature, pressure, humidity, pH, glucose, etc. This data can be used to monitor health, well-being, and quality of life. Of the articles read, 50% use measurement sensors in their solutions. Wen et al. (2021) introduces triboelectric sensors present in an intelligent glove to capture hand and finger movements during sign language. Veeralingam et al. (2019) mentions that one of the sensors can distinguish between different types of applied tension magnitudes and detect various finger movements to transmit corresponding messages to a smartphone. Gupta & Kumar (2021) mentions using an inertial measurement unit to measure hand movement through accelerometers and gyroscopes. The article of Luperto et al. (2022) introduces the concept of smart objects, where an intelligent pen with embedded sensors is used to measure user force and tremor while writing, and finally, the article of Lee, Chong & Chung (2020) discusses inertial sensors to measure acceleration, angular velocity, and the magnetic field in three distinct axes.

For detection, smart sensors can detect the presence, absence, movement, or proximity of objects, people, animals, or phenomena. This data can be used to prevent accidents, avoid obstacles, locate items or people, and activate devices or alarms. Of the articles read, 70% use measurement sensors in their solutions, making this the most used sensor category by the authors. Kim et al. (2020) detects elements present in tears to indicate the user’s health status and identify critical conditions early on. Tateno, Liu & Ou (2020) detects muscle movement through EMG signals captured by surface electrodes. The article by Veeralingam et al. (2019) describes that the gesture sensor can detect different finger gestures, such as stretching them, to indicate that the user is feeling better. The article of Luperto et al. (2022) discusses the use of microphones for user command detection, such as help requests or commands for the presented assistive system. The article of Santhalingam et al. (2020) includes word activation detection in the system. Lee, Chong & Chung (2020) mentions inertial sensors to capture movement patterns, user hand, and body position and speed. Chang, Castillo & Montes (2022) introduces early warning sensors that assess risks and prevent disasters through detection and monitoring. The device proposed by Du et al. (2022) utilizes an array of omnidirectional microphones to detect sound sources from a full 360° range. This capability allows it to identify specific speech targets amidst background noise, enhancing the clarity of audio for students with hearing loss.

For monitoring, smart sensors can continuously track the state, performance, or activity of systems, equipment, or people. This data can be used to control, optimize, correct, or improve processes, functions, or behaviours. Of the articles read, 50% use measurement sensors in their solutions. The article of Kim et al. (2020) introduces a real-time vital sign monitoring system; Liu, Jiang & Gowda (2020) and Gupta & Kumar (2021) use sensors to continuously capture finger and hand movement and position data; The article of Luperto et al. (2022) uses door sensors, accelerometers, and sensorized insoles to monitor user activity, and finally, Chang, Castillo & Montes (2022) describes the use of constant monitoring sensors to prevent disasters, in addition to using embedded systems and radio frequency communication that provides real-time information about routes and obstacles.

Smart sensors can identify and follow the location, trajectory, or destination of objects, people, or animals for tracking. This data can be used to navigate, guide, coordinate, or communicate with others. Of the articles read, 20% of them use measurement sensors in their solutions, making this the least used category by the authors. The article of Wen et al. (2021) introduces triboelectric sensors present in a glove that tracks hand and finger movements during sign language, and the Liu, Jiang & Gowda (2020) describes wearable sensors, such as rings and smartwatches, used for fine finger gesture tracking. By integrating computer vision (CV) techniques, the device proposed by Du et al. (2022) can track the position of a specific speaker. When the CV detects a human face, the system locks onto that target, ensuring that the student can hear the teacher clearly, even if the teacher moves around the classroom. In Duvernoy et al. (2023), the HaptiComm tracks the position of the user’s hand on the device allows for precise stimulation of specific areas. This tracking capability is crucial for accurately conveying tactile symbols that represent letters, words, or concepts, facilitating effective communication through tactile fingerspelling. The integration of a multi-channel data acquisition system allows for the simultaneous tracking of multiple gestures and movements Li et al. (2024), this system captures electrical signals from the hydrogel sensors attached to the user’s fingers, facilitating precise tracking of hand movements and gestures. The radar technology of Saeed et al. (2024) tracks the spatial movements of the signer, allowing for the identification of specific gestures and their sequences. This tracking functionality is crucial for understanding the context and flow of communication in sign language, as it captures the nuances of hand and body movements.

Finally, for recognition, smart sensors can recognize and interpret patterns, images, sounds, gestures, or expressions. This data can be used to interact, learn, teach, or entertain with others. Of the articles read, 50% use measurement sensors in their solutions. Articles Tateno, Liu & Ou (2020), Liu, Jiang & Gowda (2020), Gupta & Kumar (2021), Lee, Chong & Chung (2020), Du et al. (2022) and Parthasarathy et al. (2023) recognize sign language movements and gestures through wearable sensors. The article of Santhalingam et al. (2020), Saeed et al. (2024) does the same but without wearables. In hearing devices, recognizing and responding to different sounds is essential for improving auditory perception and enhancing the overall user experience. This aligns with the findings of the study, which demonstrate the effectiveness of the skin-mediated piezoelectric transduce (SPT) in transmitting sound to the cochlea and eliciting recognizable auditory responses of Furuta et al. (2022). An implantable microphone designed to enhance the hearing experience for individuals with sensorineural hearing loss by Cary et al. (2022).

The examples presented here were extracted directly from the articles and show how intelligent sensors can be applied to assistive technologies for different categories. These devices can explore and develop many other possibilities and scenarios. The important thing is that they are designed and used to promote social inclusion and autonomy for people with special needs.

In addition to the above, we noted that 30% of the articles (Kim et al., 2020; Luperto et al., 2022; Chang, Castillo & Montes, 2022) focus on security or inclusion systems, another 30% (Kim et al., 2020; Liu, Jiang & Gowda, 2020; Luperto et al., 2022) address user health monitoring systems, and 70% (Wen et al., 2021; Tateno, Liu & Ou, 2020; Veeralingam et al., 2019; Liu, Jiang & Gowda, 2020; Gupta & Kumar, 2021; Santhalingam et al., 2020; Lee, Chong & Chung, 2020), the majority, are concerned with gesture or communication recognition systems. It is important to note that some articles fall into multiple categories, so the percentages sum to more than 100%. Analyzing the presented systems, we found that 70% (Wen et al., 2021; Kim et al., 2020; Tateno, Liu & Ou, 2020; Veeralingam et al., 2019; Liu, Jiang & Gowda, 2020; Gupta & Kumar, 2021; Lee, Chong & Chung, 2020) use wearable sensors—that is, sensors positioned on some part of the user’s body—while the remaining 30% (Luperto et al., 2022; Santhalingam et al., 2020; Chang, Castillo & Montes, 2022) employ a non-wearable approach. Table 10 presents the main findings about the sensor functions.

Access availability

Intelligent sensors are devices that can capture, process, and transmit environmental information, such as sound, light, temperature, and movement. They can be applied to assistive technologies for the hearing impaired, such as hearing aids, cochlear implants, and sound alert systems. These technologies aim to improve communication, safety, and the quality of life for people with hearing impairments.

The availability and access to intelligent sensors applied to assistive technologies for the hearing impaired depend on several factors, such as cost, regulation, infrastructure, demand, and innovation. One of the challenges is the cost of intelligent sensors and assistive technologies, which can be high for many people with hearing impairments, especially in developing countries or with low incomes. Another challenge is the regulation of intelligent sensors and assistive technologies, which can be complex and vary according to the country, type of device, degree of invasiveness, and risks involved.

Table 10 Sensor functions ordered by article year.

Article	Measurement	Detection	Monitoring	Tracking	Recognition	
Veeralingam et al. (2019)	•	•				
Santhalingam et al. (2020)		•			•	
Tateno, Liu & Ou (2020)		•			•	
Liu, Jiang & Gowda (2020)			•	•	•	
Lee, Chong & Chung (2020)	•	•			•	
Gupta & Kumar (2021)	•		•		•	
Wen et al. (2021)	•			•		
Kim et al. (2020)		•	•			
Luperto et al. (2022)	•	•	•			
Chang, Castillo & Montes (2022)		•	•			
Furuta et al. (2022)					•	
Cary et al. (2022)					•	
Du et al. (2022)		•		•	•	
Parthasarathy et al. (2023)					•	
Duvernoy et al. (2023)				•		
Li et al. (2024)				•		
Saeed et al. (2024)				•	•	

Furthermore, the infrastructure required for the operation of intelligent sensors and the demand for assistive technologies, which may be low or insufficient to stimulate investment and large-scale production, especially for more specific or customized devices, are factors that can hinder access to these sensors.

Another issue to mention is the innovation of intelligent sensors and assistive technologies, which can be slow or restricted due to ethical, legal, social, or cultural issues, such as acceptance, privacy, security, or stigmatization of people with hearing impairments. To accelerate this process, the starting point is scientific research, technological development, multidisciplinary collaboration, and the active participation of people with hearing impairments in the design and evaluation of devices.

Regarding the articles in this review, the only ones that categorically explain and provide complete information about the sensors are the articles by Tateno, Liu & Ou (2020), which mentions the company Thalmic Labs; Liu, Jiang & Gowda (2020), which uses more accessible sensors, such as smartwatches and rings, and also mentions the Oura Ring product; and Lee, Chong & Chung (2020), which provides the technical specifications of the sensors, from the low-energy Bluetooth module HC-06 to the Teensy 3.6 microcontroller, stating that the entire system can be easily replicated and adapted for other gesture recognition applications using wearable sensors. According to Furuta et al. (2022), the Skin-Mediated Piezoelectric Transducer (SPT) utilizes an exposed piezoelectric material with the skin as one electrode, demonstrating comparable impedance and vibratory characteristics to conventional piezoelectric diaphragms, effectively transmitting sound to the cochlea without exerting pressure on the skin, and showing promising results in eliciting compound action potentials in response to auditory stimuli. Cary et al. (2022) provide well-documented and accessible information on intelligent sensors used in the implantable microphone, including research foundations, material specifications, technical parameters, and collaborative efforts, making it a valuable resource for advancements in hearing restoration technologies. In Du et al. (2022), the wearable device utilizes a distributed stereo microarray microphone architecture with six microelectromechanical systems (MEMS) omnidirectional microphones and integrated computer vision sensors to achieve 360° sound capture, enabling effective speech detection, tracking, and noise reduction for hearing-impaired students in dynamic classroom environments. The wearable continuous gesture-to-speech conversion system (Parthasarathy et al., 2023) utilizes a GY 521 Inertial Measurement Unit (IMU), which includes an MPU6050 accelerometer and gyroscope ensemble, to capture acceleration and angular rate values along three axes. This data is wirelessly transmitted to a Raspberry Pi 3 Model B+ via an ESP-32 Microcontroller Unit, and the entire sensor setup is integrated into a pair of gloves for a natural and mobile gesture recognition experience. The hydrogel-based mechanical sensors used in the wearable one-handed keyboard are composed of polyacrylamide (Li et al., 2024), sodium carboxymethyl cellulose, and reduced graphene oxide, exhibiting high strain sensitivity (gauge factor of 8.18), pressure sensitivity (0.3116 kPa−1), fast response time (109 ms), and multifunctionality as strain and pressure sensors and bioelectrodes, making them ideal for gesture recognition and monitoring applications. Saeed et al. (2024) utilize ultra-wideband (UWB) radar sensors, specifically the XeThru X4M03 model, to nonintrusively collect data on British Sign Language gestures, capturing distinctive motion patterns in various lighting conditions and enabling effective detection and recognition of signs.

Some articles provide information on only some of the sensors presented. Wen et al. (2021) mention that the triboelectric sensors manufactured with a layer of Ecoflex 00-30 coated in conductive fabric, in addition to using a $12,000 Agilent InfiniiVision DSO-X 3034 A oscilloscope with a standard 10MΩ probe and the Arduino MEGA 2560 to collect the triboelectric signals through a signal acquisition module. Luperto et al. (2022) discusses very common sensors, such as microphones, accelerometers, and PIR sensors, that are widely used in IoT applications and can be found in specialized electronics stores or on the Internet.

Other articles provided little information but still gave clues on how and where to find the sensors. Veeralingam et al. (2019) mentions that the sensor is a low-cost, multifunctional solution that can be used in various applications, including personal health devices, medical devices, and the Internet of Things. Therefore, the sensor may be available for use in research and development in laboratories and companies working in these areas. Gupta & Kumar (2021) uses surface electromyogram sensors (set of three triaxial accelerometers) and inertial measurement units, which are probably available for purchase in specialized electronic, medical, or research equipment stores. Santhalingam et al. (2020) mentions the radio platform structure and cites, but in a generic way, the equipment used and combined with each other that, when researched, not much information is found about. Chang, Castillo & Montes (2022) mentions that the systems were developed through three research projects in which the authors participated, suggesting that the sensors and electronic components used may have been selected based on their availability and accessibility to researchers. Finally, Duvernoy et al. (2023) only mention that twenty-five pressure sensors are placed on the volar region of the hand to detect tactile interactions.

No articles explicitly mentioned the unavailability of a particular sensor or constituent part of the system. However, Kim et al. (2020) mentions that the contact lenses, the main solution proposed by the authors, were made in the laboratory with specific moulds and materials and are not commercially available. We sumarize the results about sensor access availability in Table 11.

Table 11 Sensor access availability.

Article	Available	Partially available	Not available	Limited information	No information	
Veeralingam et al. (2019)				•		
Tateno, Liu & Ou (2020)	•					
Liu, Jiang & Gowda (2020)	•					
Lee, Chong & Chung (2020)	•					
Gupta & Kumar (2021)				•		
Wen et al. (2021)		•				
Kim et al. (2020)			•			
Luperto et al. (2022)	•					
Santhalingam et al. (2020)				•		
Chang, Castillo & Montes (2022)				•		
Furuta et al. (2022)	•					
Cary et al. (2022)	•					
Du et al. (2022)		•				
Parthasarathy et al. (2023)	•					
Duvernoy et al. (2023)				•		
Li et al. (2024)			•			
Saeed et al. (2024)	•					

Electrical consumption

Electrical energy consumption is important to consider, as it affects device performance, autonomy, and sustainability. One of the advantages is the advancement of intelligent sensor technology, which reduces the size, weight, and energy consumption of devices, thus increasing their efficiency and portability.

Another advantage is using alternative or renewable energy sources to power smart sensors and assistive technologies for the blind. For example, some devices may use solar, kinetic, or thermal energy to generate or store electrical energy. These sources can extend the battery life of devices and reduce dependence on electrical outlets or external chargers.

One challenge is balancing energy consumption with the quality of services provided by intelligent sensors and assistive technologies for the blind. For instance, some devices might use complex image or sound processing algorithms to provide detailed information about the environment, but this might require more energy and computational resources than simpler algorithms. Another example is that some devices might use wireless or Internet connections to transmit or receive data, consuming more energy than wired or local connections.

Another challenge is managing the energy consumption of intelligent sensors and assistive technologies for the blind. For example, some devices might have systems for monitoring or controlling battery levels, alerting the user about the need for recharging or shutdown. Another example is that some devices might have energy-saving or optimization modes, adjusting the operation of sensors or services based on demand or energy availability.

The article of Wen et al. (2021) is unclear about the specific consumption of the sensors used, but it mentions that the triboelectric sensors are self-powered and that the oscilloscope does the electrical signal analysis. Furthermore, it mentions using the Arduino MEGA 2560, a low-power device (approximately 75 mA at 5V). Such information suggests that the overall consumption of the sensors might be low or reduced. Kim et al. (2020) also does not provide specific information about the electrical energy consumption of the sensors in the smart contact lens. However, the article mentions that a safe and aqueous battery powers the intelligent contact lens. Additionally, the article describes that the electrochromic alarm system is designed to be compact and have limited thickness, and mentions that the consumption during the lens colour change is 27.6 microwatts, reaching an average of 28.8 microwatts with 100 colour change cycles and that the energy potential varies from 0.4V to 0.6V. Therefore, the energy consumption of the sensors may be relatively low, allowing the battery to last for a reasonable period. Lee, Chong & Chung (2020) uses sensors and modules with a low supply voltage between 3.3V and 5V, while the Teensy 3.2 microcontroller operates with a supply voltage between 1.65V and 5.5V. However, the article does not provide specific information about the electrical energy consumption of the sensors. Energy consumption can vary depending on the operating mode and system settings.

The article of Tateno, Liu & Ou (2020) presents a solution on the market called Myo Armband. The specifications state that the device houses two 3.7V, 260 mAh lithium batteries. There is no information about the sensors’ consumption or how long the battery lasts. Veeralingam et al. (2019) do not provide specific information about the electrical energy consumption of the sensor. It only describes the sensor as a low-cost and multifunctional solution that can be used in various applications, including personal health devices, medical devices, and the Internet of Things. The solutions presented in Liu, Jiang & Gowda (2020) also revolve around widely marketed products, such as the Sony SmartWatch 3 SWR50, which features a 420 mAh battery that lasts up to 96 h in stand-by according to manufacturer information and the button-shaped sensor that acts as an intelligent ring called VMU931, with hard-to-access information but indicates that it is powered by a Raspberry Pi, suggesting that consumption is reduced. Parthasarathy et al. (2023) do not provide specific details regarding the energy consumption of the sensors used in the wearable continuous gesture-to-speech conversion system. However, it mentions that a 6000 mAh power bank is used as the power supply for the Raspberry Pi, IMU, and ESP-32 modules, indicating a focus on ensuring sufficient power for the entire system’s operation.

The energy consumption of the sensors in the article by Luperto et al. (2022) may vary depending on use and configuration, considering that the sensors make up a monitoring and domestic aid system. But, in general terms, the article does not mention or provide very little information regarding energy consumption; In Gupta & Kumar (2021), the authors do not mention energy consumption, recharge time, battery specifications, or operating voltage of the sensors used in the study. The article focuses on the proposed approach for sign language recognition and the evaluation of the accuracy of the proposed sensor system. The same goes for the article by Santhalingam et al. (2020), which does not mention electrical energy consumption. However, the system is designed to be used in home assistant devices, which are usually powered by an external power source. Additionally, the system uses a programmable 60 GHz radio transceiver and a phased antenna array, which consume less energy than other antennas. In Chang, Castillo & Montes (2022), Furuta et al. (2022), Cary et al. (2022), Du et al. (2022), Duvernoy et al. (2023), Li et al. (2024) and Saeed et al. (2024), there is no specific information about the electrical energy consumption of the sensors used in the described systems. However, the article mentions the need to develop a battery-assisted and clean energy-independent communication infrastructure from the current telecommunications system to allow more excellent reliability, which may suggest that energy efficiency may be an essential consideration in the development of these systems. Table 12 presents the main finds about the energy consumption of sensors.

Table 12 Energy consumption of sensors.

Article	High	Moderate	Low	No or little information	
Veeralingam et al. (2019)				•	
Santhalingam et al. (2020)				•	
Lee, Chong & Chung (2020)			•		
Kim et al. (2020)			•		
Tateno, Liu & Ou (2020)				•	
Santhalingam et al. (2020)				•	
Liu, Jiang & Gowda (2020)			•		
Gupta & Kumar (2021)				•	
Wen et al. (2021)			•		
Luperto et al. (2022)				•	
Chang, Castillo & Montes (2022)				•	
Furuta et al. (2022)				•	
Cary et al. (2022)				•	
Du et al. (2022)				•	
Parthasarathy et al. (2023)				•	
Duvernoy et al. (2023)				•	
Li et al. (2024)				•	
Saeed et al. (2024)				•	

Communication protocols

Communication protocols are sets of rules and standards that allow different devices or systems to communicate with each other efficiently and securely. They define aspects such as the form, content, format, speed, and sequence of exchanged messages. Some examples of communication protocols are TCP/IP, HTTP, Bluetooth, Wi-Fi, and NFC.

Some examples of communication protocols include TCP/IP (Transmission Control Protocol/Internet Protocol), which is the most widely used protocol on the Internet and allows communication between computers and devices connected to the global network. It comprises four layers: application, transport, Internet, and network. It ensures data delivery in order and without errors. HTTP (Hypertext Transfer Protocol) is an application protocol allowing data transfer between web browsers and servers. It uses TCP/IP as a transport protocol and defines methods for requesting and sending information, such as GET, POST, PUT, and DELETE. Bluetooth is a wireless communication protocol that connects nearby devices, such as cell phones, headphones, keyboards, mice, printers, etc. It uses radio-frequency waves in the 2.4 GHz range and has low power consumption. Wi-Fi (Wireless Fidelity) is a wireless communication protocol that allows the connection between devices on a local network or the Internet. It uses radio-frequency waves in the 2.4 GHz or 5 GHz range and has high transmission speed, near field communication (NFC) is a wireless communication protocol that allows data exchange between very close devices, such as cell phones, cards, tags, etc. It uses radio-frequency waves in the 13.56 MHz range and has a low transmission speed.

Communication protocols applied to smart sensors aim to facilitate the interaction of these devices with the world around them, providing them with more functionality, efficiency, and security. However, these protocols also present some challenges and limitations, such as the need for compatibility between the devices or systems involved in communication, dependence on energy sources, such as batteries or outlets, the possibility of failures, and accessibility.

Wen et al. (2021) introduce triboelectric sensors that are connected to an Arduino MEGA 2560 with an IoT module, which, in turn, is connected to a computer that receives and recognizes the signals through a deep learning algorithm in Python. This computer passes the information to the augmented reality environment built with Unity through TCP/IP and LAN protocols.

Tateno, Liu & Ou (2020) do not explicitly mention the communication protocols of the Myo armband sensor. However, according to the manufacturer’s website, the Myo armband uses Bluetooth 4.0 to communicate with external devices, such as computers, smartphones, and tablets. The armband is also compatible with a variety of development platforms, including Windows, Mac, iOS, and Android, and offers an API for developers who want to create custom applications that use the EMG data collected by the armband. The article Veeralingam et al. (2019) also uses a Bluetooth module to communicate with the mobile application through a connection with a microcontroller, thus performing the transmission. The mobile application is specialized and was developed to analyze the sensor data, process them, and display them in real-time;

The research of Liu, Jiang & Gowda (2020) mentions that the VMU931 sensor (button-shaped sensor used on the index finger as a ring) is connected to a Raspberry Pi via a USB cable to transmit data, as it does not support wireless transmission. The Raspberry, in turn, transmits the data via WiFi. However, the article does not provide additional information about the communication protocols used by the smartwatch, for example. Researching the manufacturer’s website, it is known that the device has a USB input, integrated Wi-Fi, NFC, and Bluetooth, but it is not possible to determine which of these protocols the authors used. The Gupta & Kumar (2021) mentions that the signals are transmitted wirelessly to a receiving device connected to a computer via USB. The signals are then recorded and later processed for sign language recognition. The sensors used in the study consist of surface electromyogram sensors and inertial measurement units, which measure information related to linear acceleration, orientation, and rotation rate of the users’ hand and arm movements.

The sensors mentioned in the article Luperto et al. (2022) communicate wirelessly with a hub providing a cloud gateway. The adopted communication protocol is MQTT, which is a lightweight messaging protocol designed for devices with limited bandwidth and unstable connections. It uses a publish/subscribe model to send messages between devices connected to the Internet.

Santhalingam et al. (2020) state that the NI+SiBeam platform is connected to a host that sends/receives data to/from FPGA and implements additional signal processing tasks. In mmASL, the Tx and Rx hosts are connected via Ethernet for control and coordination (e.g., setting a specific Tx and Rx beam sector during scanning).

The IMU sensors and the Teensy 3.2 microcontroller in the article by Lee, Chong & Chung (2020) communicate using the I2C (Inter-Integrated Circuit) protocol, which is a short-distance synchronous serial communication protocol. To allow the connection of all six IMU sensors to the microcontroller, two TCA9548 multiplexers are used as communication means. Each multiplexer can connect up to four IMU sensors and has four pairs of serial data (SDA) and serial clock (SCL) pins. The article also mentions that the system transmits the sensor data to a terminal via a low-energy HC-06 Bluetooth module, which operates on the BLE (Bluetooth Low Energy) 4.0 protocol.

Chang, Castillo & Montes (2022) describe several systems and solutions that use different communication protocols for different functions. For example, the early warning system for natural disasters uses a battery-assisted, clean, energy-independent communication system from the current telecommunications system, which may include different communication protocols, such as Wi-Fi and Bluetooth. The communication system to improve the mobility of visually impaired people in public transport and buildings uses radio frequency (RF) communication to provide real-time information about routes and obstacles. However, the article does not provide a complete list of all the communication protocols used in each system.

Kim et al. (2020) do not provide enough information to infer the communication protocol used. The article mentions that previous studies used wireless communication to transmit real-time data measured in tears. Electrical stimuli may be sent through a cable connected to the intelligent contact lens.

Parthasarathy et al. (2023) employs Bluetooth Low Energy (BLE) for wireless communication between the sensors and the Raspberry Pi. The GY 521 IMU, which includes the accelerometer and gyroscope, transmits data wirelessly to the ESP-32 Microcontroller Unit (MCU), which in turn communicates with the Raspberry Pi. This setup allows for efficient data transfer while maintaining low power consumption, suitable for a wearable device.

The communication protocol employed by Li et al. (2024) the wearable one-handed keyboard involves a wireless transmission system using Bluetooth. Specifically, the HC08 Bluetooth module is utilized to wirelessly transmit the resistance values resulting from the deformation of the hydrogel sensors to a smartphone, enabling real-time data communication and interaction. This setup allows for efficient data transfer and enhances the usability of the wearable device.

Saeed et al. (2024) mentions the use of the Extensible Messaging and Presence Protocol (XEP) for configuring the radar sensors. This protocol is employed through the x4driver interface, allowing for communication and control of the sensor during the data collection process. However, the text does not provide information about other communication protocols that may have been used. For additional details on other protocols.

In the research presented in Furuta et al. (2022), Cary et al. (2022), Du et al. (2022), Duvernoy et al. (2023), the authors do not specify the exact communication protocols used for the sensors.

We have summarized the results of the communications protocols in Table 13.

Table 13 Communication protocols.

Article	Bluetooth	BLE	I2C	LAN	MQTT	RF	TCP/IP	USB	Wi-Fi	XEP	Unspecified Wireless	No Information	
Veeralingam et al. (2019)	•												
Santhalingam et al. (2020)							•						
Kim et al. (2020)												•	
Tateno, Liu & Ou (2020)	•												
Liu, Jiang & Gowda (2020)								•	•				
Lee, Chong & Chung (2020)		•	•										
Gupta & Kumar (2021)								•			•		
Wen et al. (2021)				•			•						
Luperto et al. (2022)					•				•				
Chang, Castillo & Montes (2022)	•					•			•				
Furuta et al. (2022)												•	
Cary et al. (2022)												•	
Du et al. (2022)												•	
Duvernoy et al. (2023)												•	
Parthasarathy et al. (2023)		•											
Li et al. (2024)	•												
Saeed et al. (2024)										•			

Physical proportions

The physical proportions of smart sensors depend on their type, purpose, and place of use. Some factors that can influence the size and shape of the sensors are sensitivity, accuracy, autonomy, durability, ergonomics, aesthetics, safety, and cost. Intelligent sensors should generally be small, lightweight, discreet, comfortable, durable and reliable. However, there are cases where more extensive or visible sensors may be preferable, such as when you want to draw attention to a particular function or facilitate interaction with the device.

The physical proportions of intelligent sensors are the relative dimensions between the parts that make up the sensor, such as the sensor element, the electronic circuit, the power source, the communication interface, etc. These proportions can vary according to the design and functionality of the sensor. For example, a pressure sensor may have a very small sensor element but a relatively sizeable electronic circuit to process and transmit data. On the other hand, an image sensor may have a significant sensor element, but a compact electronic circuit is needed to integrate the capture and processing of images.

Wen et al. (2021) present a table with the areas of the triboelectric sensors used in the intelligent glove in units of square centimetres (cm2). The areas range from 0.25 cm2 to 5.25 cm2, depending on the sensor’s position in the glove. However, the article does not provide information on other physical proportions of the sensors, such as their thickness or shape. Considering the glove, it is understood that the sensors have more reduced proportions.

Kim et al. (2020) do not provide specific information about the physical proportions of the sensor. However, the article describes that the electrochromic alarm system is designed to be compact and have limited thickness, allowing it to be integrated into a smart contact lens. The article also mentions that the system was able to maintain contact with tears due to its curved shape. By describing an intelligent contact lens, it is inferred that the proportions are reduced.

In Tateno, Liu & Ou (2020), it is mentioned that the Myo armband contains eight surface electromyography (EMG) sensors and is worn on the arm to measure the EMG signals generated by the arm muscles during hand movements. Looking for the armband specifications, the following proportions are found: 11.94 × 7.37 × 10.41 cm and 249 g. Thus, it is understood that the physical proportions of the sensors are small.

Veeralingam et al. (2019) do not provide specific information about the physical proportions of the sensor. However, it describes that the sensor is based on NiSe2 grown on cellulose paper using the hydrothermal solution processing method and that the overall cost of the sensor is about 0.3 dollars. From the images provided by the author, it is noted that the sensors are positioned next to the fingers and hands, making it possible to infer that the proportions of the sensors are reduced.

Liu, Jiang & Gowda (2020) does not provide specific information about the sensors’ physical proportions. The article describes the use of a VMU931 sensor, which is described as “robust and compact”, but does not provide additional information about its physical dimensions. However, it is known that the wearable sensors used are a watch and a ring. The ring is relatively compact, as seen in the image provided by the author. The watch, looking for specifications, has the following dimensions: 51 ×36 × 10 mm and weighs 76 g, so it is known that the dimensions are small.

Gupta & Kumar (2021) used a sensor system consisting of three sEMG sensors and two six-degree-of-freedom IMU sensors, which were placed on both forearms of users in a bracelet configuration. Through the images, it is possible to see that the sensors are small and positioned on the forearms, occupying a small space.

Luperto et al. (2022) does not provide specific information about the physical proportions of the sensors. It describes the types of sensors used in the system, such as thin accelerometers, microphones, PIR sensors, and switching sensors, but does not provide details about their physical dimensions. The physical dimensions of the sensors may vary depending on the model and manufacturer. The Giraff-X robot has a height comparable to that of humans, 1.70 m. It has a display at the top and a two-wheel differential drive system with two swivel wheels that allow it to turn on the spot. The study’s main product is the robot, which has elevated dimensions, and considering that several other small and medium-sized sensors make up this monitoring system, this set will be characterized by significant proportions.

Santhalingam et al. (2020) does not provide specific information about the physical proportions of the mmASL and the sensors. However, the article provides an image of the assembled system, indicating that it has large proportions, given the radio platform used and the other components of medium proportions. The whole set has large proportions.

Lee, Chong & Chung (2020) does not provide specific information about the physical proportions of the sensors. However, the work presents a table that summarizes the components of the sensing module, including the IMU sensors, the Teensy 3.2 microcontroller, the TCA29548A multiplexer, and the HC-06 Bluetooth module. The table provides information about the technical specifications of each component, such as the operating voltage, the clock frequency, and the transmission range. However, there is no information about the physical dimensions of the sensors, but it is known from the image provided in the article that the sensors are positioned on the hand, like a glove, making it possible to infer that the proportions are reduced.

Chang, Castillo & Montes (2022) does not provide specific information about the physical proportions of the sensors used in the described systems. The article’s main focus is describing the systems developed to improve the inclusion of people with hearing and visual disabilities in early warning systems for natural disasters, mobility in public transport and buildings, and information systems with a monitoring panel. The article provides detailed information about the electronic components and communication protocols used in each system but does not provide specific information about the physical proportions of the sensors. From the images, it seems that the sensors do not have large proportions, but the system itself is quite extensive.

The dimensions of the sensors employed in the Skin-Mediated Piezoelectric Transducer (Furuta et al., 2022) are a thickness of 0.12 mm and a diameter of 10.0 mm for the piezoelectric material, while the metal electrode plate has a thickness of 0.1 mm and a diameter of 15.0 mm.

Parthasarathy et al. (2023) specifies that the GY 521 IMU, which includes the MPU6050 accelerometer and gyroscope, is designed to be compact, with the entire system being described as having a size equivalent to a credit card. This compact design enhances the mobility of the wearable device. However, specific dimensions or proportions of the individual sensors themselves are not detailed in the text.

In Li et al. (2024), the physical dimensions of the sensors used in the one-handed wearable keyboard are 10 mm × 10 mm × 1 mm. These dimensions make the sensors compact and suitable for applications in wearable devices.

Saeed et al. (2024) do not provide specific physical dimensions of the radar sensors used. It mentions the use of the XeThru X4M03 model for data collection, but detailed specifications regarding its physical dimensions are not included in the text.

Du et al. (2022), Cary et al. (2022) and Duvernoy et al. (2023) do not specify the physical dimensions of the sensors used in the wearable device. But, we can infer a proportion from the images of the system, as show in Table 14.

Table 14 Sensors’ physical proportions.

Article	Small	Medium	Large	No information	
Veeralingam et al. (2019)	•				
Kim et al. (2020)	•				
Lee, Chong & Chung (2020)	•				
Tateno, Liu & Ou (2020)	•				
Liu, Jiang & Gowda (2020)	•				
Santhalingam et al. (2020)			•		
Wen et al. (2021)	•				
Gupta & Kumar (2021)	•				
Chang, Castillo & Montes (2022)	•				
Luperto et al. (2022)			•		
Furuta et al. (2022)	•				
Du et al. (2022)	•				
Cary et al. (2022)	•				
Duvernoy et al. (2023)	•				
Parthasarathy et al. (2023)	•				
Saeed et al. (2024)				•	
Li et al. (2024)	•				

Table 14 shows the sensors’ physical proportions of the selected articles.

Sensor combination

The combination of intelligent sensors in assistive technologies represents a significant advancement in the field of accessibility and in improving the quality of life for people with physical or cognitive disabilities. These sensors are designed to collect data from the environment and the user, process them in real time, and provide appropriate responses or assistance. One of the main advantages of this approach is customization. By combining multiple sensors, it is possible to create solutions highly tailored to the individual needs of each user. This allows assistive technologies to be more effective and efficient in performing specific tasks, such as manipulating objects, locomotion, or communication.

Another significant advantage is the synergy between sensors. Combining different types of sensors, such as cameras, proximity sensors, accelerometers, and microphones, allows for a more comprehensive understanding of the environment and user actions. This improves the accuracy and reliability of assistive technologies, making them more intuitive to use. For example, a system combining vision and motion sensors can help a visually impaired person navigate unknown spaces more safely and effectively.

However, using combined sensors in assistive technologies can have challenges and disadvantages. One of the main challenges is integrating and synchronizing different sensors, which can be complex and require a high level of technical expertise. Additionally, combining sensors can increase assistive technologies’ development and maintenance costs, making them less accessible to some people who need them.

Another potential disadvantage is the issue of data privacy and security. Collecting sensitive information through sensors can raise concerns regarding the storage and use of this data. To protect users’ integrity and privacy, it is essential to implement robust security measures and ensure compliance with privacy regulations.

The article from Wen et al. (2021), introduces a glove designed to recognize sign language gestures using the combination of signals from multiple sensors, suggesting that the individual use of a single sensor may not be sufficient to recognize gestures accurately. Additionally, the article mentions that the glove has 15 sensors in different positions, each with a customized coverage area, suggesting that combining multiple sensors is crucial for gesture recognition.

In Kim et al. (2020), the system is standalone and does not require external equipment to operate. However, the article also mentions that previous studies used wireless communication to transmit real-time data measured in tears, suggesting that other types of sensors or measurement equipment might be used in conjunction with smart contact lenses to collect and transmit information. Initially, the system operates independently.

Tateno, Liu & Ou (2020) describes that the Myo armband can be used individually. The article describes how the Myo armband was used to collect EMG data from individuals while performing sign language gestures. The EMG data was then processed and used to control a virtual reality system, providing users with visual and auditory feedback. The system was tested on a group of 20 participants, and the gestures performed by the participants could be accurately recognized. Therefore, the Myo armband sensor can be used individually to collect EMG data and control assistive technology systems.

The sensor can be employed individually in the article Veeralingam et al. (2019). The sensor is a “smart multifunctional sensor” that can be customized for a specific type of detection using a user-friendly Android app. The app can be accessed remotely via a smartphone, allowing the user to monitor the sensor data conveniently and accessibly. Additionally, the article suggests that the sensor can be used for personal health monitoring and diagnosing various diseases, implying that the sensor can be used individually.

The authors in Liu, Jiang & Gowda (2020) describe using a VMU931 sensor connected to a Raspberry Pi via a USB cable to transmit data. Therefore, the sensor cannot be used individually; it requires an external device to process and transmit the data. However, it describes using IMU sensors (inertial measurement unit) in an intelligent ring and a smartwatch to track finger movement. These sensors can be used individually, but the complete system described in the study requires the use of both sensors to track finger movement accurately.

In Gupta & Kumar (2021), it is mentioned that the sensor system consists of three sEMG sensors and two IMU sensors, which were placed on both forearms of users in a bracelet configuration. The signals are transmitted via a wireless network to a receiving device, which passes the information to a computer through a USB cable. Furthermore, the sensors used in the study are made up of surface electromyogram sensors and inertial measurement units, suggesting that to ensure the expected accuracy, they should be used in combination.

Luperto et al. (2022) state that each sensor can be employed individually. For example, the door sensor can be used to monitor whether the user enters or exits the house, while the thin accelerometers can be used to monitor the user’s sleep and rest behavior. Microphones can detect user commands, such as help requests or commands for the robot, while the main switch can turn the entire system on and off. Each sensor has a specific function and can be used individually or in conjunction with other sensors, depending on the user’s needs.

Santhalingam et al. (2020) does not provide specific information on whether the mmASL sensor can be employed individually. However, the article mentions that the mmASL is designed for home assistant devices, suggesting that the system is designed to be integrated into existing devices. Additionally, the article mentions that the mmASL is implemented using a programmable 60 GHz radio system with a phased antenna array from SiBeam and multiple FPGA platforms from National Instruments. This suggests that the mmASL is a complex system that might be challenging to implement individually. However, without additional information, it is impossible to provide specific information on whether the mmASL sensor can be employed individually.

Lee, Chong & Chung (2020) does not provide specific information on the possibility of employing sensors individually. However, it describes that the sensing module consists of several components, including IMU sensors, a Teensy 3.2 microcontroller, a TCA29548A multiplexer, and an HC-06 Bluetooth module. These components are integrated into a wearable sign language gesture recognition system. It is possible that the IMU sensors can be used individually in other contexts, but this would depend on the technical specifications and the needs of the system in question. Therefore, the sensors here depend on each other for full functionality.

In Chang, Castillo & Montes (2022), the authors describe that sensors can be employed individually in different applications, depending on the specific needs of each case. For example, the article describes the use of temperature and humidity sensors to monitor environmental conditions in buildings and prevent mould formation and other health problems. Additionally, the article describes the use of motion and proximity sensors to improve the mobility of visually impaired people in indoor and outdoor environments. In general, sensors can be used in a wide variety of applications, from environmental monitoring to motion detection and physiological parameter measurement, depending on the specific needs of each case.

The Skin-Mediated Piezoelectric Transducer used in Furuta et al. (2022) can potentially be employed using different combinations of sensors, once it efficiently transmits sound to the cochlea by skipping the tympanic membrane and middle ear ossicles, indicating versatility in its application. Different sensor combinations could enhance its functionality or adapt it for various clinical settings, although specific configurations are not detailed by authors.

The nature of radar technology and deep learning employed by Saeed et al. (2024) could potentially be integrated into networks or used in combination with other sensors to enhance data collection and improve the accuracy of sign language detection.

The proposed wearable device by Du et al. (2022) can potentially be combined with other sensors or used in sensor networks. The article discusses the integration of a microphone array and computer vision techniques, indicating that the system is designed to enhance speech tracking and noise reduction for hearing-impaired students. This flexibility suggests that it could be integrated with additional sensors to improve functionality or used as part of a larger network of devices to provide comprehensive support in various environments, such as classrooms.

Cary et al. (2022) do not explicitly mention whether the implantable microphone can be combined with other sensors or used in sensor networks. However, the design of such devices typically allows for integration with other sensors to enhance functionality, such as improving sound localization or providing additional auditory information. In general, implantable devices can be designed to work with other sensors, potentially forming a network that could improve the overall performance of assistive hearing devices. Further research or additional studies would be needed for specific details on integration capabilities.

Duvernoy et al. (2023) do not explicitly mention whether the HaptiComm can be combined with other sensors or used in sensor networks, but its desired flexibility and modularity suggest that such integration could be a possibility.

The wearable continuous gesture-to-speech conversion system by Parthasarathy et al. (2023) can potentially be combined with other sensors or used in sensor networks. The document discusses the use of a GY 521 IMU for gesture recognition, which is a versatile sensor that can be integrated with additional sensors to enhance functionality. For example, combining it with sensors for environmental monitoring, heart rate monitoring, or other biometric data could provide a more comprehensive system. Moreover, the wireless communication capabilities (via Bluetooth Low Energy) of the ESP-32 module allow for integration into larger sensor networks, where multiple devices can communicate and share data. This could facilitate applications in smart environments or assistive technologies, where multiple sensors work together to provide richer data and improved user experiences.

In Li et al. (2024), the hydrogel-based mechanical sensors used in the one-handed wearable keyboard can be combined with other sensors and utilized in sensor networks. The study highlights the potential for integrating these sensors into a broader system for motion tracking and human-machine interfaces. Additionally, the flexibility and compatibility of the hydrogel sensors allow for their incorporation with various types of sensors, enhancing the functionality and application range of wearable devices. This capability supports the development of more complex and multifunctional sensor networks for diverse applications. Table 15 presents an overview of the sensor combination analysis utilized in the selected articles.

Table 15 Sensor combination.

Reference	Can be employed individually	Cannot be employed individually	No information	
Veeralingam et al. (2019)	•			
Santhalingam et al. (2020)		•		
Liu, Jiang & Gowda (2020)		•		
Lee, Chong & Chung (2020)		•		
Tateno, Liu & Ou (2020)	•			
Gupta & Kumar (2021)		•		
Wen et al. (2021)		•		
Kim et al. (2020)	•			
Luperto et al. (2022)	•			
Chang, Castillo & Montes (2022)	•			
Furuta et al. (2022)			•	
Du et al. (2022)			•	
Cary et al. (2022)			•	
Duvernoy et al. (2023)			•	
Parthasarathy et al. (2023)			•	
Saeed et al. (2024)			•	
Li et al. (2024)			•	

Methodological challenges and gaps

Methodological challenges and gaps in scientific articles are aspects that can compromise the quality, validity, and reliability of research conducted in different areas of knowledge. These aspects involve issues such as defining the research problem, choosing the most appropriate methods and techniques to collect and analyze data, interpreting and communicating results, and ethics and integrity in conducting and disseminating research, among others. Overcoming these challenges and filling these gaps is fundamental to the progress of science, as it allows for the advancement of knowledge, problem-solving, innovation, and social development.

Wen et al. (2021) mentions the use of various measuring equipment for the acquisition and processing of signals generated by triboelectric sensors in the smart glove. This equipment includes a charge amplifier, an analog-digital converter, a microcontroller for signal acquisition, and signal processing software for data analysis. The system was evaluated in an augmented reality environment, where users could communicate via text and audio. The results showed that the system can recognize sign language with high accuracy and that communication through the system is effective and intuitive. The article also discusses the system’s limitations, such as the need for sensor calibration and the lack of support for other sign languages. Overall, the article presents a promising solution for communication between people with hearing impairments and those unfamiliar with sign language, suggesting possibilities for future research and developments in the area.

Kim et al. (2020) introduces a system capable of monitoring vital signs and alerting the user in critical conditions, such as hypoglycemia or hyperglycemia. It can also provide additional visual information to the user, such as real-time captions or navigation information. The article presents an intelligent contact lens capable of producing cocolor changes in response to electrical stimuli. It was tested on an animal model and could detect and alert about increased intraocular pressure, a common glaucoma symptom. The article discusses the system’s potential as a non-invasive, low-cost ocular health monitoring tool that can be easily integrated into patients’ daily routines. Furthermore, the article suggests that the system can be adapted to detect and alert about other medical conditions, such as diabetes and high blood pressure. The article concludes that the smart contact lens with an electrochromic alarm system is a promising technology for ocular health monitoring and may have applications in other health areas.

The research presented by Tateno, Liu & Ou (2020) selected a set of sign language movements from the ASL library for recognition and used a bilinear model to reduce individual differences between people. The results showed that the system was effective in recognizing sign language movements. It was concluded that the proposed system, which uses the Myo armband to control a virtual reality system, is viable and can provide visual and auditory feedback to deaf or hearing-impaired individuals. The system accurately recognized the gestures performed by participants and provided real-time feedback. However, the article also highlights some system limitations, including improving gesture recognition accuracy and testing the system on a more extensive and more diverse group of participants. Additionally, the article discusses the importance of considering users’ individual needs and preferences when designing assistive technology systems.

Veeralingam et al. (2019) discusses manufacturing a low-cost multifunctional sensor based on NiSe2 grown on cellulose paper using the hydrothermal solution processing method. The sensor can detect breathing rate, pH, and finger movements and can be customized for a specific type of detection using a user-friendly Android app. The article discusses the sensor’s features, such as its ability to withstand 300 to 500 bending or tension cycles, and suggests areas for future research and improvements, such as testing the sensor in more acidic or alkaline pH solutions. The article concludes that the sensor is a promising solution for personal health monitoring and diagnosing various diseases and can be used in various applications, including wearable electronics, electronic skin devices, and the Internet of Things.

On the other way, Liu, Jiang & Gowda (2020) offers comprehensive conclusions and discussions based on the study’s findings. It assesses the accuracy of finger spelling gesture detection using the proposed system, comparing the results with those of similar studies. The study also highlights its limitations, such as the small sample size and lack of participant diversity. Additionally, it examines the potential to enhance gesture detection accuracy through the use of visual or auditory signals. The article considers the possibility of employing glove-based sensors instead of those in rings or watches for sign language recognition. Moreover, it explores the application of neural networks to further improve detection accuracy. In conclusion, while the proposed system demonstrates promise in detecting sign language gestures, significant challenges remain in achieving greater precision and accessibility.

In Gupta & Kumar (2021), the authors present an innovative approach to sign language recognition using wireless sensors, which can help overcome the limitations of other camera and data glove-based approaches. The study highlights the importance of collecting data from users with different sign language skill levels and developing robust and accurate sign recognition models. The results show that the proposed sensor system can achieve a sign recognition rate of up to 96.5% for users with intermediate sign language skills. Furthermore, it emphasizes the importance of exploring different sensor combinations and signal processing techniques to further improve sign recognition accuracy.

In turn, Luperto et al. (2022) discusses the limitations and lessons learned during the field experimental campaign with the Giraff-X robot and the AAL platform. Some of the limitations include the lack of control over the usage environment, difficulty recruiting participants, and the need for frequent system maintenance. Some of the lessons learned include the importance of training for elderly users, the need to simplify system installation and use, and the importance of providing adequate technical support. The article also discusses the importance of collecting long-term usage data to better understand the acceptability and feasibility of assistance robots for the elderly. Moreover, the study’s main conclusion is that social assistance robots can be successfully used for long-term elderly care. The robot setup improved the system’s overall effectiveness, allowing participants to use its assistive features for longer and collecting more meaningful monitoring data about them. However, the robot’s presence harmed the participants’ system evaluation, indicating that the acceptability of SAR-based systems is not yet fully established. The study suggests that future work should investigate how to improve interactions between the elderly and SARs, especially in the long term, and how to deal with the initial novelty effect that may fade over time.

Santhalingam et al. (2020) introduce mmASL, a sign language recognition system based on 60 GHz millimeter-wave signals. This system is designed for home assistant devices to help deaf or hearing-impaired users communicate through sign language. The article discusses the challenges associated with sign language recognition using radio signal-based sensors, such as the presence of other people in the environment and the need for high accuracy and low latency. It also presents a detailed experimental analysis of mmASL’s performance in different test scenarios, including the presence of other people in the environment. The experimental results show that mmASL can recognize sign language gestures with high accuracy and tolerance to the presence of other people in the environment. The article’s conclusions suggest that mmASL is a promising technology for improving communication for deaf or hearing-impaired users in home environments.

Lee, Chong & Chung (2020) provides an in-depth discussion on the development and evaluation of a wearable sign language gesture recognition system based on a deep learning model. The article addresses the challenges associated with creating a system capable of accurately recognizing sign language gestures in real time. It presents a comprehensive experimental analysis of the system’s performance across various test scenarios, including environments with multiple people and varying lighting conditions. The results demonstrate that the proposed system achieves high accuracy and low latency in gesture recognition. The article concludes by highlighting the potential of this system as a promising technology to enhance communication for deaf or hearing-impaired users in diverse settings.

In Cary et al. (2022), the authors discuss several methodological challenges and gaps in the development of the implantable microphone for assistive hearing devices, including the need to achieve a high signal-to-noise ratio (SNR) critical for performance in noisy environments, the unwanted detection of body noise that can affect sound clarity, and the requirement for irreversible surgical procedures for implantation, which may deter potential users. Additionally, the lack of natural directionality provided by the ear’s anatomy in current external microphones hinders sound localization and understanding in complex auditory environments. The article also emphasizes the importance of experimental validation of the microphone’s performance, indicating that further testing and refinement are necessary to optimize the design, suggesting that while the technology shows promise, significant hurdles remain in terms of performance, usability, and integration with existing systems.

Du et al. (2022) identify several key challenges in the development of assistive devices for hearing-impaired students, with particular emphasis on the absence of suitable solutions that do not require teachers to wear transmitters, which can be cumbersome. The authors underscore the necessity of miniaturization, enhanced intelligence, and personalization in hearing aids, alongside the critical importance of improving sound source identification (SOI) in complex noise environments. Furthermore, they suggest that future research should prioritize the optimization of microphone array designs and investigate various types of microphones to enhance detection performance and address the limitations of current systems.

For Furuta et al. (2022), one major concern is the potential for electrical injury to the skin when using it as an electrode, necessitating further evaluation of the minimum electrical current that could cause harm. Additionally, the difference in skin impedance between humans and rodents poses a challenge, as the effects observed in rodent models may not directly translate to human applications. The authors emphasize the need for clinical evaluations to assess the SPT’s effectiveness and safety in human subjects, considering the differences in histology and physiological properties between human and rodent skin.

In Parthasarathy et al. (2023), the authors identify several methodological challenges and gaps in their gesture-to-speech conversion system. Notably, they highlight difficulties in accurately recognizing continuous gestures and differentiating them from non-gestural movements, which can result in miscommunication. The system’s limited gesture vocabulary further constrains its usability, underscoring the need to expand gesture recognition capabilities. Additionally, the authors emphasize the importance of developing effective data augmentation techniques to enrich the training dataset and improve system robustness. Achieving real-time gesture processing while maintaining high accuracy is another significant challenge, and the integration of the system with existing IoT frameworks for broader applications remains a critical avenue for future research and development.

Duvernoy et al. (2023) highlight several scientific challenges related to the HaptiComm device, particularly the need for further research to optimize actuation timing and speed parameters, as well as to evaluate the device’s performance in recognizing individual and sequenced letters compared to human fingerspelling. The authors also stress the importance of quantifying the user learning curve to facilitate the effective adoption of this assistive technology. Moreover, they underscore the need for the device to transmit tactile information in a clear and natural manner, while maintaining ergonomic flexibility to accommodate different hand sizes and user preferences.

In Saeed et al. (2024), the authors highlight challenges such as the need for non-intrusive methods that do not rely on wearable devices or complex camera setups, which can be cumbersome and affected by environmental factors. Additionally, they emphasize the difficulty in accurately interpreting the intricate spatiotemporal patterns of sign language gestures using radar data, necessitating the development of advanced deep learning models tailored for this application. These challenges underscore the need for further exploration and innovation in the field of sign language detection using radar technology.

In the development of hydrogel-based wearable sensors for human-machine interaction presented in Li et al. (2024), one major gap is the limited exploration of interactive wearables specifically designed for individuals with single-arm functionality or disabilities, which hinders accessibility and usability. Additionally, the authors highlight challenges in achieving high accuracy and responsiveness in gesture recognition, particularly in real-time applications. There is also a need for improved feature identification methods to enhance the performance of machine learning algorithms used in gesture recognition. Furthermore, the integration of these sensors into larger sensor networks poses challenges in terms of compatibility and data management, necessitating further research and development to optimize their functionality in practical applications.

In conclusion, the articles discussed in this section present various technological solutions for improving communication for deaf or hearing-impaired users. These solutions range from wearable sensors and intelligent gloves to smart contact lenses and robots. While these technologies show promise, challenges remain to be addressed, such as improving accuracy, reducing latency, and ensuring user acceptance. However, with continued research and development, these technologies have the potential to significantly improve the quality of life for deaf or hearing-impaired individuals.

Conclusions

The esteemed systematic literature review aimed to organize and present the current state of intelligent sensors in assistive technologies so that the deaf can use this information for a larger purpose in the future. From the selected articles, several scientific solutions were identified for the development of devices to assist deaf individuals. It was observed that these solutions extend beyond hearing aids and can be combined to form new integrated solutions: the variety of sensors found indicates a wide range of possibilities to assist this population in leading better lives through these technologies.

Thanks to the methodology employed, it was possible to conceive an organized chronology of topics to be included in this research, making all the material available on a comprehensive and accessible online platform. Although some articles did not provide all the specifications, it is still possible to understand the scenario of the most commonly used sensors to assist people with hearing impairments. As mentioned above, some details were missing, such as information regarding maintenance, availability, and accessibility of the sensor, as well as its consumption and cost.

Furthermore, it is essential to emphasize that these solutions go beyond hearing aids and can be combined to form new integrated solutions. The systems developed by the authors are promising and can be integrated at different stages to create a more extensive system that can assist in disaster risk events, monitoring systems, early detection, security systems, and other risk situations that may occur in daily routines. The systems developed by the authors are especially important for aiding or assisting people with hearing and visual impairments, among others.

The results obtained so far confirm several previously stated hypotheses, such as the fact that the proposed inclusive systems allowed for the evaluation, prevention, and mitigation of risks. The systems developed by the authors range from broader approaches of inclusive intelligent cities, aiming to improve the quality of life for all individuals regardless of their abilities or limitations, to more specific situations, like a home assistant filled with sensors and monitoring devices.

In conclusion, interdisciplinary collaboration and end-user involvement in developing these inclusive and accessible systems are crucial for the success of these technologies. In summary, the systematic literature review aimed to organize and present the current state of intelligent sensors in assistive technologies so that the deaf can use this information for a larger purpose in the future.

Additional Information and Declarations

Competing Interests

Author Contributions

Data Availability

The authors declare there are no competing interests.

Caio César Sabino Soares conceived and designed the experiments, performed the experiments, analyzed the data, performed the computation work, prepared figures and/or tables, authored or reviewed drafts of the article, and approved the final draft.

Luis Augusto Silva conceived and designed the experiments, performed the experiments, analyzed the data, performed the computation work, prepared figures and/or tables, authored or reviewed drafts of the article, and approved the final draft.

Anita Fernandes conceived and designed the experiments, performed the experiments, analyzed the data, authored or reviewed drafts of the article, and approved the final draft.

Gabriel Villarrubia González analyzed the data, authored or reviewed drafts of the article, and approved the final draft.

Valderi R.Q. Leithardt analyzed the data, authored or reviewed drafts of the article, and approved the final draft.

Wemerson Delcio Parreira conceived and designed the experiments, performed the experiments, analyzed the data, performed the computation work, authored or reviewed drafts of the article, and approved the final draft.

The following information was supplied regarding data availability:

This is a literature review.

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
