# Peer review of "Intelligent sensors in assistive systems for deaf people: a comprehensive review"

_PeerJ Computer Science, doi:10.7717/peerj-cs.2411_

## Round 0.1 · original submission · Minor Revisions

To address the reviewers' comments when resubmitting the manuscript, the authors should:

- Correct citation formatting, e.g., revise citations to ensure proper format throughout the paper.
- Strengthen the rationale for focusing on sensors.
- Separate the discussion of sign and spoken language aspects, as combining them may confuse readers.
- Include the time frame of the studies reviewed in the abstract, which is currently missing.
- Review and integrate the latest research and technologies in the field to improve the methodology.

Reviewer 1 ·

Basic reporting

The article is clear and ambiguous but in some parts the citations are not written in a correct format. eg.
588 The (Wen et al., 2021) presents a table.....


There are sufficient literature review but the reason for focusing on sensors is a little weak. Also the audio and visual aspects seem to be put in one place so it may be easier to separate the language aspect to the environmental sound aspect.

Experimental design

The methodology for collecting the articles are clear and the use of tables and figures help the reader understand the study design. Yet the definition of sensor seem a little vague and the reason for focusing on sensors remain still weak.

Validity of the findings

The findings are interesting but as mentioned in the basic reporting, the sign language aspect and spoken language aspect should be separated. Mixing the two will make it difficult for the deaf to find the appropriate data or findings that suit their needs.

Additional comments

Overall interesting topic and the data collection provides insights into sensors that can aid the deaf. It would be more interesting to add some social background of the deaf and the different sensors that can aid different types of deaf or hard of hearing people. As for the technology side the paper thoroughly covers the main topic.

Cite this review as

Reviewer 2 ·

Basic reporting

Good

Experimental design

The abstract does not mention the time frame for the studies reviewed. This could be a limitation if the review only covers recent work or omits older but relevant studies.

sources adequately cited


The review organized logically into coherent paragraphs/subsections

Methodology can improve based on latest technologies.

Validity of the findings

The systematic literature review successfully organizes and presents the current state of intelligent sensors in assistive technologies, specifically for the deaf. The review identifies various scientific solutions and discusses their potential beyond traditional hearing aids, which supports the broader goal of enhancing the quality of life for deaf individuals. The argument is further supported by references to specific systems, such as inclusive intelligent cities and home assistants, indicating a comprehensive analysis of how these technologies can be applied in real-world situations.

The conclusion identifies unresolved questions, gaps, and future directions. It notes that certain articles did not provide complete specifications, such as information on sensor maintenance, availability, accessibility, consumption, and cost. Additionally, it highlights the need for further research into integrating these solutions into more comprehensive systems that can handle disaster risk events and other daily challenges. The conclusion also emphasizes the importance of interdisciplinary collaboration and end-user involvement, pointing to future directions for improving the development and acceptance of these technologies.

Additional comments

Methodology can improve based on latest technologies. So study latest research works in this filed.

Cite this review as

---

## Round 0.2 · accepted · Accept

I confirm that the new version has been improved and the authors have addressed the reviewers' comments, which were largely minor.